# Northern Hemisphere Snow-Cover Trends (1967–2018): A Comparison between Climate Models and Observations

**Ronan Connolly [1,2,*], Michael Connolly [2], Willie Soon [1], David R. Legates [3], Rodolfo Gustavo Cionco [4] and Víctor. M. Velasco Herrera [5]**

[1]   Center for Environmental Research and Earth Sciences (CERES), Salem, MA 01970, USA; 2018ceres@gmail.com
[2]   Independent Scientist, Dublin, Ireland
[3]   College of Earth, Ocean, and the Environment, University of Delaware, Newark, DE 19716-2541, USA; legates@udel.edu
[4]   Comisión de Investigaciones Científicas de la Provincia de Buenos Aires—Grupo de Estudios Ambientales, Universidad Tecnológica Nacional, Colón 332, San Nicolás 2900, Buenos Aires, Argentina; gcionco@frsn.utn.edu.ar
[5]   Instituto de Geofisica, Universidad Nacional Autónoma de México, Ciudad Universitaria, Coyoacán 04510, México D.F., Mexico; vmv@geofisica.unam.mx
*   Correspondence: ronanconnolly@yahoo.ie

**Abstract:** Observed changes in Northern Hemisphere snow cover from satellite records were compared to those predicted by all available Coupled Model Intercomparison Project Phase 5 ("CMIP5") climate models over the duration of the satellite's records, i.e., 1967–2018. A total of 196 climate model runs were analyzed (taken from 24 climate models). Separate analyses were conducted for the annual averages and for each of the seasons (winter, spring, summer, and autumn/fall). A longer record (1922–2018) for the spring season which combines ground-based measurements with satellite measurements was also compared to the model outputs. The climate models were found to poorly explain the observed trends. While the models suggest snow cover should have steadily decreased for all four seasons, only spring and summer exhibited a long-term decrease, and the pattern of the observed decreases for these seasons was quite different from the modelled predictions. Moreover, the observed trends for autumn and winter suggest a long-term increase, although these trends were not statistically significant. Possible explanations for the poor performance of the climate models are discussed.

**Keywords:** Northern Hemisphere snow cover; CMIP5 climate models; climate change

## 1. Introduction

Snow cover represents one of the main components of the cryosphere, along with sea ice [1], permafrost [2], and the various ice sheets and glaciers [3,4]. Seasonal snow cover also represents a major component of the hydrological cycle in mid- and high-latitudes [5]. Snow cover also supports a large winter outdoor recreation industry, while snowmelt is an important source of water for many societies. As a result, Sturm et al. [6] estimate that the financial value of snow to human society is of the order of trillions of dollars. Boelman et al. [7] further stress that understanding changes and trends in snow cover is also important for the study of wildlife communities of ecosystems that experience seasonal snow.

Temporal changes in snow cover are an important part of global climate change for at least two reasons. First, total snow cover is widely considered a key indicator of climate change [5–10]. Climate models from the 1970s to the present have consistently predicted that human-caused global warming from increasing atmospheric greenhouse gas concentrations should be causing a significant and continual decline in total snow cover [8–18]. Second, changes in snow cover can further contribute to global climate change by altering the Earth's surface albedo (i.e., the fraction of incoming sunlight reflected to space), and also because snow cover partially insulates the ground below it [8–14,16–21].

Weekly satellite-derived observations for the Northern Hemisphere are available from November 1966 to the present (historical snow cover data provide less spatial and temporal coverage for the Southern Hemisphere). These estimates are a collaborative effort between the National Oceanic and Atmospheric Administration (NOAA) and the Rutgers University Global Snow Lab [9,10,19,22–27]. This dataset (henceforth, termed the "Rutgers dataset") represents the longest running satellite-based record of any environmental variable, and it is widely used by the climate science community [9,10,19,22–27].

Various ground-based measurements of local snowfall and snow cover extend back prior to the pre-satellite era. Brown and Robinson [28] were able to combine these data sources with the Rutgers dataset to extend estimates of Northern Hemisphere snow cover for March and April back to 1922 (and to 1915 for North America). By averaging the two monthly estimates, they derived a combined "spring" estimate. Along with the Rutgers dataset, this second observational dataset will be considered briefly in this paper.

In the 1970s and early 1980s, trends in satellite-derived estimates of Northern Hemisphere snow cover were a cause for consternation within the scientific community. Although climate models had predicted that global (and hemispheric) snow cover should have decreased from human-caused "global warming" due to the increasing atmospheric carbon dioxide concentrations [8], Northern Hemisphere snow cover had actually *increased* since at least the start of the record. At the time, this led to some skepticism about the validity of the climate models, (e.g., [9,10,24]).

In the late-1980s, average snow cover finally began to decrease. Although Robinson and Dewey [10] cautioned that it was still too "premature to infer an anthropogenic cause for the recent decrease in hemispheric snow cover", this reversal of trends provided a renewed confidence in the climate models and the human-caused global warming theory (which was by now generating considerable interest from the public).

Still, as time passed, it became increasingly apparent that the observed changes in snow cover were quite different from what the models had predicted. While the models had predicted declines for all four seasons, the observed decrease in snow cover was largely confined to the spring and summer months, and not the autumn/fall or winter [24–26].

Moreover, the decrease in spring and summer largely occurred in a single step-change decrease in the late-1980s. That is, the spring and summer averages remained relatively constant until the late-1980s, then dropped during the late-1980s, and have remained relatively constant at that lower value since [24–26], although a further decrease in summer extent appears to have occurred over the last decade [26]. The climate model-predicted decrease from human-induced warming consisted of a continuous trend rather than a single step-change. Although Foster et al. [25] were careful not to rule out the possibility that some of the decrease might "be at least partially explained by human-induced warming" [26] (p. 155), they argued that this step-like change seemed more consistent with a late-1980s shift in the Arctic Oscillation, or some other climatic regime shift.

Nonetheless, several studies noted that when calculating a linear trend for the observed spring (or summer) values over a time period that covered the late 1980s step-change, e.g., 1967–2012, the decline introduced a "negative trend", and that the continual decline predicted by the climate models also implied a "negative trend" for those seasons (albeit, also for the other seasons). Moreover, since the observed decrease occurred for both spring and summer, the annual averages also implied a net negative trend [11,13–18,29,30]. Also, it was noted that the climate models did at least qualitatively

replicate the overall annual cycle, i.e., the fact that snow cover increases in the autumn and winter and decreases in the spring and summer [11,14,16].

By comparing linear trends for the spring over a fixed time-period, it could be argued that some agreement existed between the climate model predictions and the observations [11,13–18,29,30]. In particular, the widely-cited Intergovernmental Panel on Climate Change's (IPCC) Working Group 1 Fifth Assessment Report [29] used the observed negative 1967–2012 trend in Northern Hemisphere spring snow cover extent as one of its main arguments for concluding that the global warming since the 1950s was very unusual, "Warming of the climate system is unequivocal, and since the 1950s, many of the observed changes are unprecedented over decades to millennia. The atmosphere and ocean have warmed, *the amounts of snow and ice have diminished*, sea level has risen, and the concentrations of greenhouse gases have increased" ([29], emphasis added in italics) (p. 2).

Some studies have gone further and used climate model-based "detection and attribution" studies to argue that the decline in spring snow cover cannot be explained in terms of natural variability and must be due to human-induced global warming [15,17]. Essentially, these studies compared the output of the Coupled Model Intercomparison Project Phase 5 (CMIP5) climate models [31] using "natural forcings only" and those using "natural and anthropogenic forcings" to the observed spring trends and were only able to simulate a negative trend using the latter [15,17]. This is similar to the approach the IPCC 5th Assessment Report used for concluding that, "It is extremely likely that human influence has been the dominant cause of the observed warming since the mid-20th century." [28] (p. 15). Soon et al. [32], however, showed that the CMIP5 climate modelling groups only considered a small subset of the available estimates of solar variability for their "natural forcings" and that if other (equally plausible) estimates had been considered, much (or possibly all) of the post-1950s warming could have been explained in terms of natural climate change.

Focusing on linear trends still highlighted puzzling discrepancies between the model predictions and observations. First, the observed negative linear trends for spring are considerably larger in magnitude than the model-predicted trends for spring [11,14–18,30]. Although widely-noted, this does not itself appear to have generated much criticism of the reliability of the climate models—perhaps because the signs of the trends are in this case the same. A second discrepancy is that snow cover increased for all seasons in the Tibetan Plateau region [33–35], and more generally, China [35], while the climate models predicted this region should have experienced a decrease in snow cover.

However, a third major discrepancy is more controversial and has led to considerable debate. In recent years, many regions in the Northern Hemisphere have experienced heavy snow storms in the autumn or winter (e.g., the winters of 2009/2010 and 2010/2011). Partly for this reason, linear trends for autumn and winter suggest an increase in snow cover [16,24,36–38], or at least that snow cover has remained reasonably constant [39]. This is in sharp contrast to the climate models which (as explained above) have predicted a continual decrease for all four seasons.

Räisänen [12] argued that the response of the total winter snow fall to global warming is non-trivial because increases in temperature tend to increase precipitation, and if the air temperature remains low enough for the precipitation to be in the form of snow, this can lead to an increase in the total snow fall. Under a period of warming, the models predict an increase in snowfall for some regions (where the average temperature is below about −20 °C), but a decrease in other regions [12,13]. However, the regions where the models predict an increase in snowfall are regions that are already snow-covered. Therefore, this cannot explain the observed increase in winter snow cover.

Cohen et al. [36] suggested that the climate models might be missing important climatic processes—specifically, that a decrease in summer Arctic sea ice could contribute to the observed increase in October snow cover for Eurasia, and that this, in turn, alters the Arctic Oscillation, leading to colder winters (and a greater winter snow cover). Liu et al. [37] have made similar arguments. Brown and Derksen [40] argue that a bias might exist in the Eurasian October snow cover data, and that the large increasing trend for October is simply an artefact of bad observations. While Mudryk et al. [16] offer support to this argument, they note that the problem largely was confined to October in the

Eurasian region. Moreover, in a follow-up to the Cohen et al. study which specifically avoided the use of the October trends, Furtado et al. [41] presented further evidence that the CMIP5 climate models systematically fail to "capture well the observed snow [–Arctic Oscillation] relationship". More recently, Hanna et al. [42] have argued that the CMIP5 climate models also fail to capture the summer Greenland high-pressure blocking phenomenon.

Although several studies have compared the observed spring snow cover trends to climate model predictions [11,14–18,30], little direct comparison of trends for other seasons has been conducted [16]. Moreover, most of the comparisons have focused exclusively on linear trends, while the observed trends often show distinctly non-linear fluctuations from year to year. Therefore, in this paper, we directly compare the observed Northern Hemisphere snow-cover trends for all four seasons (and annual trends) to the CMIP5 climate model hindcasted trends. Our analysis compares both the linear trends (obtained by linear least squares fitting) and the time-series themselves.

## 2. Materials and Methods

Monthly snow cover data for the Northern Hemisphere (in square kilometers) were downloaded from the Rutgers University Global Snow Lab website, https://climate.rutgers.edu/, in January 2019. This dataset begins in November 1966 and provides an almost complete time-series up to January 2019, although some months in the early portion of the record (i.e., July 1968, June–October 1969, and July–September 1971) have no data. We estimated the values for these missing months as the mean of the equivalent monthly values in the year before and after the missing year.

Global Climate Model output from the CMIP5 model runs were obtained from the Koninklijk Nederlands Meteorologisch Instituut's (KNMI's) Climate Explorer website, https://climexp.knmi.nl/, in January 2019 using a Northern Hemisphere land mask. Monthly Northern Hemisphere snow cover from this source were reported as either a percentage or a fraction of the total land area (depending on the format used by each modelling group). We converted these values into square kilometers by multiplying by the total Northern Hemisphere land area ($1.00281 \times 10^8$ km$^2$).

For our comparisons with the Rutgers dataset, we were interested in the modelled output over the same period for which we had observations, i.e., 1967–2018. Each of the submitted CMIP5 model runs covered a much longer period, i.e., 1861–2100. The first part of each run (1861–2005) was generated using "historical forcings". However, the rest of the run (2006–2100) used one of four representative concentration pathway (RCP) scenarios [43]: "RCP2.6", "RCP4.5", "RCP6.0", and "RCP8.5". These scenarios offer different projections of how greenhouse concentrations could increase to 2100 and are named according to the estimated extra "radiative forcing" (in W/m$^2$) this is projected to have added by 2100. Although these scenarios substantially diverge during the 21st century, the differences between the projections by 2018 are relatively modest. For example, if we consider the 1967–2016 annual trend (a metric which will be discussed in Section 3.1), the multi-model mean using all 196 model runs (across all four scenarios) is $-29,800 \pm 1600$ km$^2$/year. Meanwhile, if we calculate the multi-model means separately for each scenario, the results are: RCP2.6 = $-30,600 \pm 3600$ km$^2$/year; RCP4.5 = $-30,200 \pm 2500$ km$^2$/year; RCP6.0 = $-26,500 \pm 3400$ km$^2$/year; and RCP8.5 = $-30,700 \pm 3200$ km$^2$/year. In other words, the trends up to 2016 are comparable across all scenarios. On the other hand, as will be discussed in Section 3.1, the exact "internal variability" differs for each scenario run. For this reason, we treated each of the scenario runs as a separate run.

As can be seen from Table 1, 15 modelling groups (from 9 countries) contributed snow cover estimates to the CMIP5 project as part of their model output. Some modelling groups used more than one model version (and some tested different "physics versions") and/or multiple runs for each of the four scenarios. We noted that the Climate Explorer website did not provide snow cover results for some of the CMIP5 modelling groups (e.g., the UK's Hadley Centre) even though the website provides output for these models for other parameters, e.g., surface air temperature. It is possible that some of these missing models either did not calculate or did not submit snow cover output to the CMIP5 project. At any rate, we analyzed output from 196 model runs (from 24 climate models).

**Table 1.** All Coupled Model Intercomparison Project Phase 5 (CMIP5) climate model runs used for the analysis in this article. These corresponded to the models that provided snow cover estimates.

| Modelling Group | Country | Model Name | RCP2.6 | RCP4.5 | RCP6.0 | RCP8.5 | Total |
|---|---|---|---|---|---|---|---|
| Beijing Climate Center | China | bcc-csm1-1 | 1 | 1 | 0 | 1 | 3 |
|  |  | bcc-csm1-1-m | 1 | 1 | 1 | 0 | 3 |
| Beijing Normal University | China | BNU-ESM | 1 | 1 | 0 | 1 | 3 |
| Canadian Centre for Climate Modelling and Analysis | Canada | CanESM2 | 5 | 5 | 0 | 5 | 15 |
| National Center for Atmospheric Research | USA | CCSM4 | 6 | 6 | 6 | 6 | 24 |
| Community Earth System Model Contributors | USA | CESM1-BGC | 0 | 1 | 0 | 1 | 2 |
|  |  | CESM1-CAM5 | 2 | 2 | 2 | 3 | 9 |
| Centre National de Recherches Météorologiques | France | CNRM-CM5 | 1 | 1 | 0 | 5 | 7 |
| Commonwealth Scientific and Industrial Research Organisation | Australia | CSIRO-Mk3-6-0 | 10 | 10 | 10 | 10 | 40 |
| Laboratory of Numerical Modelling for Atmospheric Sciences and Geophysical Fluid Dynamics (LASG) | China | FGOALS-g2 | 1 | 1 | 0 | 1 | 3 |
| First Institute of Oceanography | China | FIO-ESM | 3 | 3 | 3 | 3 | 12 |
| NASA Goddard Institute for Space Studies | USA | GISS-E2-H | 3 | 15 | 3 | 3 | 24 |
|  |  | GISS-E2-H-CC | 0 | 1 | 0 | 0 | 1 |
|  |  | GISS-E2-R | 0 | 1 | 0 | 0 | 1 |
|  |  | GISS-E2-R-CC | 0 | 1 | 0 | 0 | 1 |
| Institute for Numerical Mathematics | Russia | inmcm4 | 0 | 1 | 0 | 1 | 2 |
| Japan Agency for Marine–Earth Science and Technology | Japan | MIROC5 | 3 | 3 | 3 | 3 | 12 |
|  |  | MIROC-ESM | 1 | 1 | 1 | 1 | 4 |
|  |  | MIROC-ESM-CHEM | 1 | 1 | 1 | 1 | 4 |
| Max Planck Institute | Germany | MPI-ESM-LR | 3 | 3 | 0 | 3 | 9 |
|  |  | MPI-ESM-MR | 1 | 3 | 0 | 1 | 5 |
| Meteorological Research Institute | Japan | MRI-CGCM3 | 1 | 1 | 1 | 1 | 4 |
| Norwegian Climate Centre | Norway | NorESM1-M | 1 | 1 | 1 | 1 | 4 |
|  |  | NorESM1-ME | 1 | 1 | 1 | 1 | 4 |
| 15 modelling groups | 9 countries | 24 models | 46 | 65 | 33 | 52 | 196 |

Rather than studying the trends for all twelve months separately, we considered just the four Northern Hemisphere seasons: winter (December, January, and February – henceforth DJF), spring (March, April, and May – henceforth MAM), summer (June, July, and August – henceforth JJA), and autumn/fall (September, October, and November – henceforth SON). The winter averages for a given year used the December from the preceding calendar year. We also discuss the annual averages (i.e., January to December). However, Brown and Robinson's [28] "spring" dataset consists of the average over only two months (March and April). For our comparison with the Brown and Robinson [28] dataset, we used March/April averages only.

The Brown and Robinson [28] dataset was downloaded from the supplementary information provided with their paper. However, their time-series excludes Greenland (due to a shortage of pre-satellite era measurements) and only extends to 2010. As can be seen from Figure 1a,

the relationship between the two time-series was highly linear ($r^2$ = 0.994). Therefore, for comparison with the climate model output, we rescaled their time-series so their values were commensurate with those of the Rutgers dataset for the period of overlap (i.e., 1967–2010), and the uncertainty envelope provided with the dataset (95% confidence interval) was rescaled accordingly. The Brown and Robinson series was then updated with the Rutgers March/April averages (2011–2018)—see Figure 1b. Error bars for the updated period were assumed to have the same values as the average of the error bars for the final five years of the Brown and Robinson series (i.e., 2006–2010).

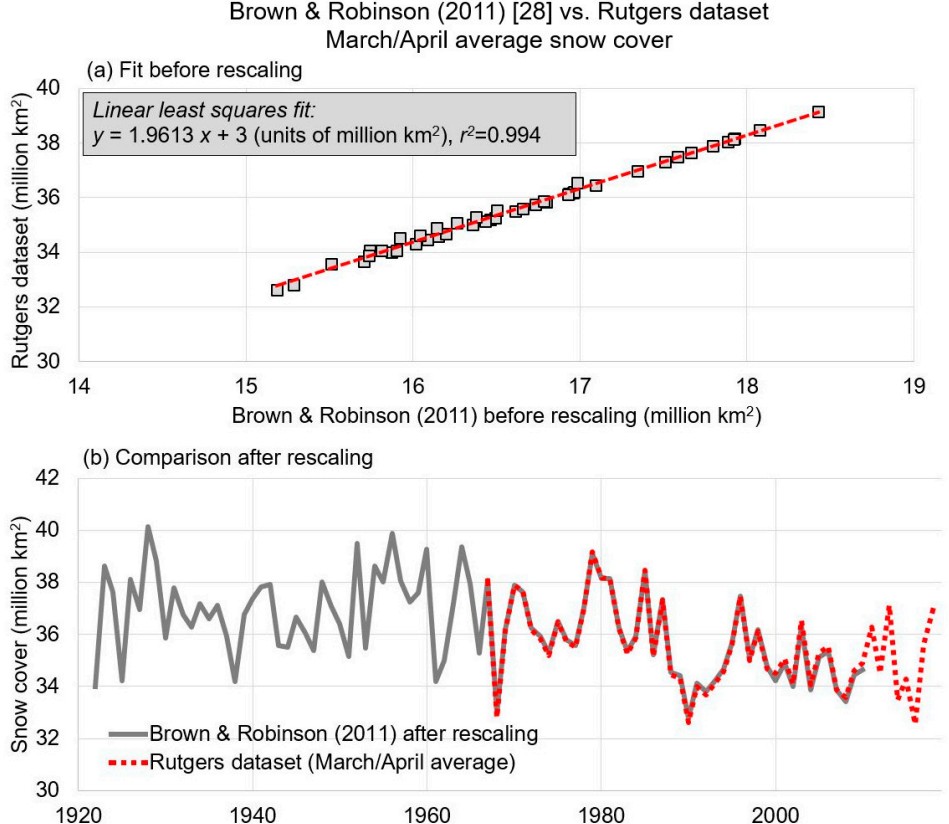

**Figure 1.** (**a**) Relationship between the Brown and Robinson [28] dataset (*x*-axis) and the Rutgers dataset (*y*-axis) for the average March/April snow cover over the period of common overlap (1967–2010). (**b**) Comparison of the rescaled version of Brown and Robinson [28] with the Rutgers dataset for average March/April snow cover.

## 3. Results

### 3.1. Comparison of CMIP5 Climate Modelled Snow-Cover Trends to the Satellite-Derived Rutgers Dataset

Observed Northern Hemisphere snow cover (km$^2$) for all four seasons (Figure 2a–d) and the annual average (e) to the equivalent values for each of the 196 CMIP5 runs, averaged over the 50-year period (1967–2016) is shown. The models tended to underestimate the observed values for all four seasons, and a wide range existed for all seasons (although smallest for summer). However, the models did seem to at least capture the general annual cycle in that the snow cover reached a maximum in winter and a minimum in summer, with intermediate values for the spring and autumn/fall. This qualitative replication of the annual cycle was noted by others and suggests that there is at least some realism to the models [11,14,16].

Because each model run implies a different average snow cover and these values are typically smaller than the observed averages, a direct comparison between the absolute trends can be challenging.

Thus, for the rest of the paper, each time-series is converted into an anomaly time-series relative to the 1967–2016 average.

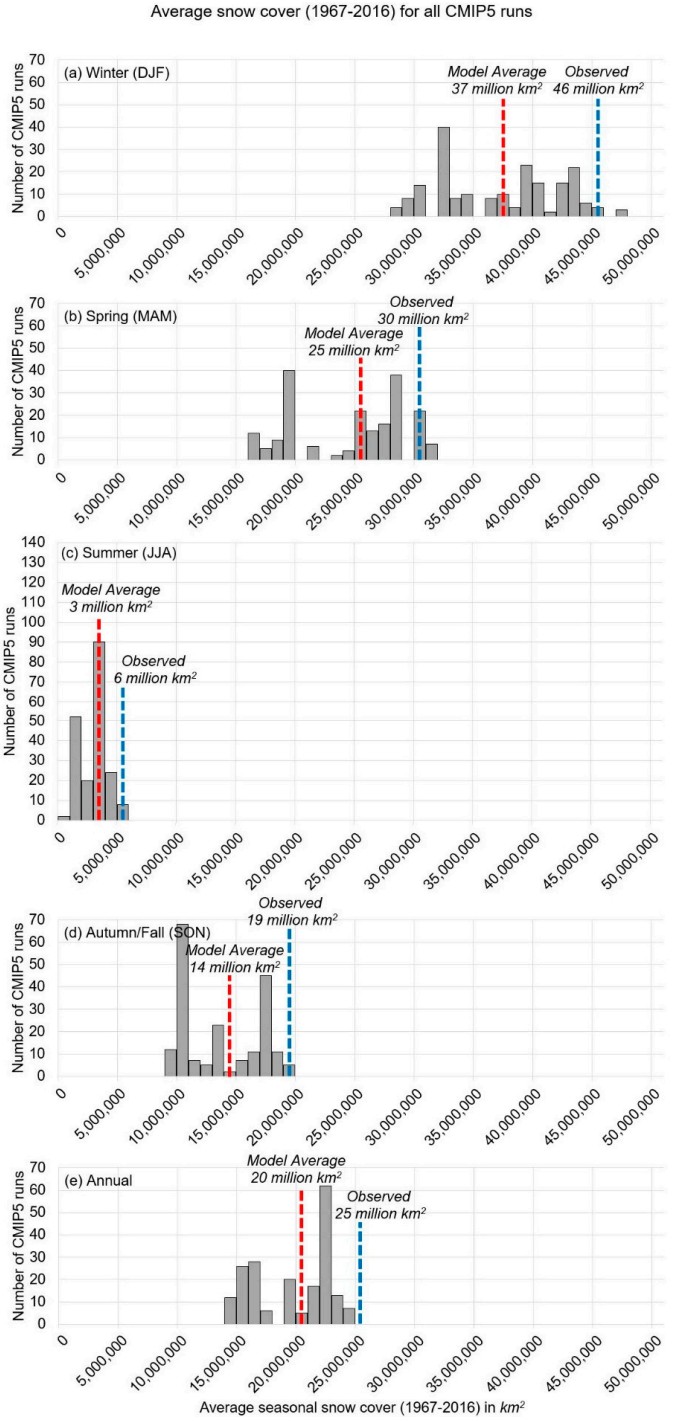

**Figure 2.** Estimated 1967–2016 Northern Hemisphere snow cover area for each of the 196 CMIP5 simulations for (**a**) winter, (**b**) spring, (**c**) summer, (**d**) autumn/fall, and (**e**) annual averages. The observed areas for each season (or yearly average) are indicated with dashed, blue lines in each panel.

The observed 1967–2016 linear trends were compared to the equivalent trends of each of the 196 CMIP5 runs for each season (Figure 3). Although the observed annual linear was consistent with that of the models (Figure 3e), we can see that this was due to the fact that the models significantly underestimated the negative summer trend (Figure 3c), and to a lesser extent, that of spring (Figure 3b),

while failing to predict the positive trends for winter (Figure 3a) and autumn (Figure 3d). In other words, the models poorly described trends for three of the four seasons (winter, summer, and autumn), and fared little better at describing trends in the spring.

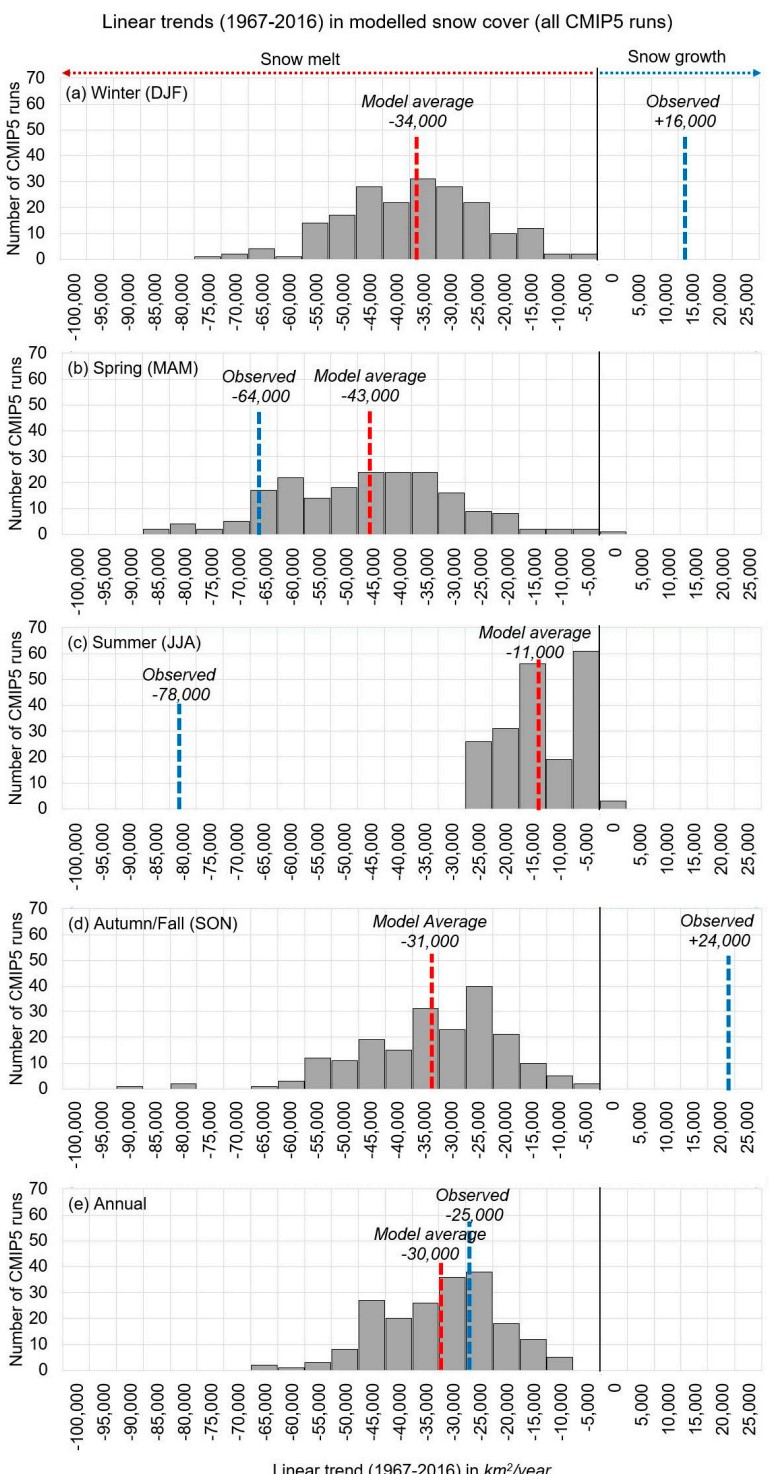

**Figure 3.** Distribution of linear 1967–2016 Northern Hemisphere snow cover trends for all 196 CMIP5 simulations for (**a**) winter, (**b**) spring, (**c**) summer, (**d**) autumn/fall, and (**e**) annual averages. The observed trends for each season (or yearly average) are indicated with dashed, blue lines in each panel.

As discussed in the introduction, it can be misleading to limit a comparison of climate models and observations to the linear trends over a single time-period, since the observed time-series were not linear in nature. Technically, a linear trend can be nominally computed for any interval, but if the time-series is non-linear in nature, then this can misleadingly imply a "linear" nature to the data which is absent.

To remedy this, the entire time-series was plotted (Figure 4a) for observed annual snow cover (relative to the 1967–2016 mean). Annual snow cover was lower after the mid-1980s relative to what it was before the mid-1980s. Thus, the linear trend *implies* a long-term decrease of $-25,000$ km$^2$/year, but this was largely an artefact of the step-like drop in the mid-1980s [25,26]. Indeed, the last two years had above-average snow cover.

The multi-model means of all 196 CMIP5 runs (Figure 4b) showed that unlike the observations, the model-predicted trends were reasonably well described in terms of a decreasing linear trend ($-30,000$ km$^2$/year). Qualitatively, this can be seen by visually comparing the two plots. The observations plot (Figure 4a) showed a considerable amount of yearly variability, while the multi-model mean (Figure 4b) showed a gradual but almost continuous decline from 1967 to the present.

While the linear fit associated with the multi-model mean had an $r^2$ of 0.93, that associated with the observations was only 0.19. Due to the long time-period, all linear fits were statistically significant ($p = 0.0014$ for the observations and $p = 10^{-28}$ for the multi-model mean). Also, the error bars (uncertainty) associated with the linear fits were much greater for the observations ($\pm 15,000$ km$^2$/year) than that for the multi-model mean ($\pm 2000$ km$^2$/year).

On this basis, the current climate models appear to be unable to explain the observed trends and are therefore inadequate. However, one might disagree because the multi-model mean is largely determined by the "external forcings" that are input into the models and does not reflect the "internal variability" of individual model runs.

With the current climate models, global snow cover is largely dictated by global temperatures (hence they predict that global snow cover should decrease due to the predicted human-induced global warming from greenhouse gases). If a simulation run is adequately equilibrated and not majorly affected by "drift", then the global temperatures for a given year are mostly determined by:

1. External radiative forcing from "anthropogenic factors". This includes many factors, but atmospheric greenhouse gas and stratospheric aerosol concentrations are the main components.
2. External radiative forcing from "natural factors". Currently, models consider only two: changes in total solar irradiance ("solar") and stratospheric aerosols from volcanic eruptions ("volcanic").
3. Internal variability. This is the year-to-year random fluctuations in a given model run. As we will discuss below, some argue that this can be treated as an analogue for natural climatic inter-annual variability.

As Soon et al. [32] noted, the CMIP5 models only consider a small subset of the available total solar irradiance estimates, and each of the estimates in that particular subset implied that solar output has been relatively constant since the mid-20th century (perhaps with a slight decrease). Meanwhile, the "internal variability" of each model yields different random fluctuations (since they are random). Therefore, the internal variability of the models tends to cancel each other in the multi-model mean.

Thus, the 1967–2018 trends of the multi-model mean are almost entirely determined by the modelled "anthropogenic forcing" (a net "human-induced global warming" from increasing greenhouse gases) and short-term cooling "natural forcing" events from the two stratospheric volcanic eruptions that occurred over that period (i.e., the El Chichón eruption in 1982 and the Mount Pinatubo eruption in 1991). However, clearly, the observed trends in annual snow cover (Figure 4a) are more complex than that relatively simple explanation.

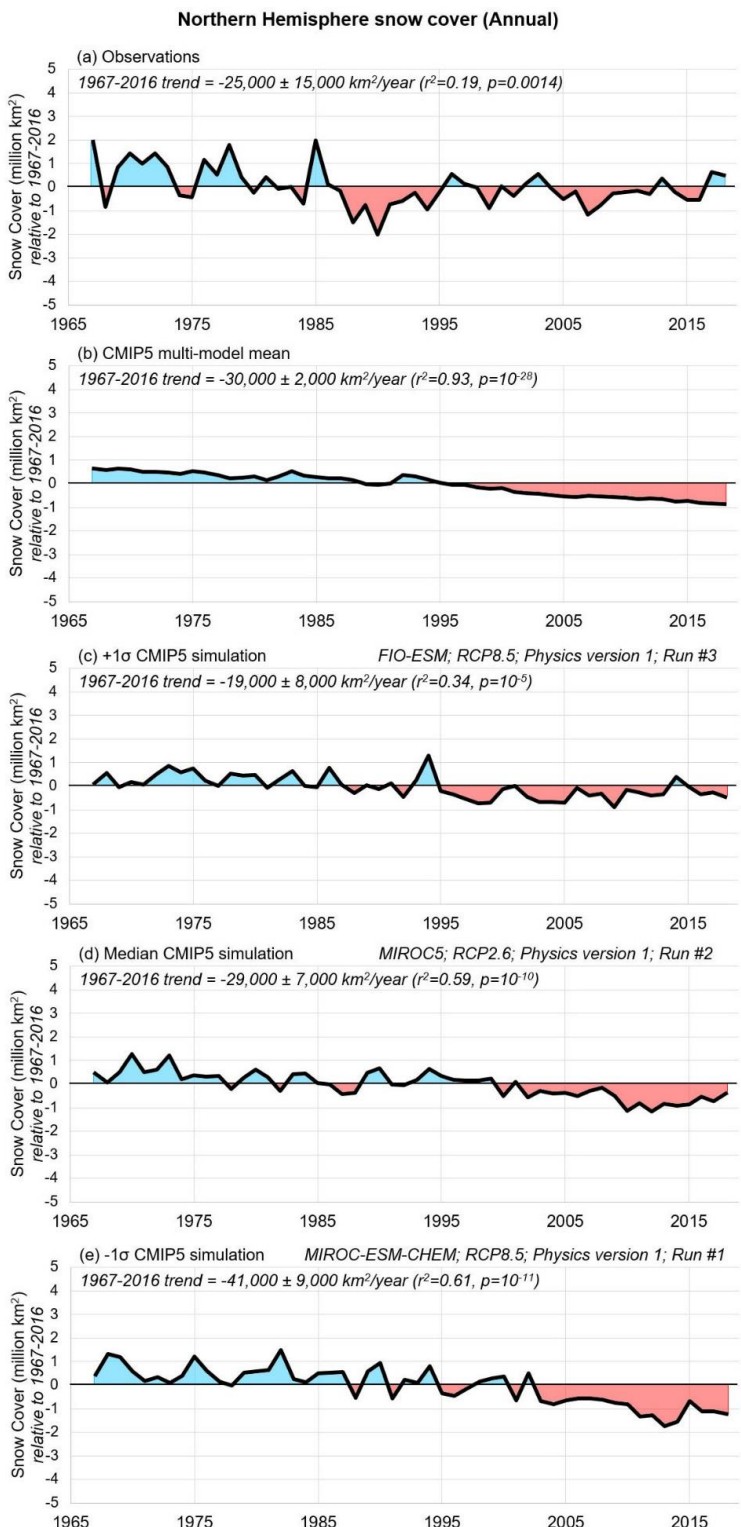

**Figure 4.** Annually-averaged trends in Northern Hemisphere snow cover (relative to 1967–2016) according to (**a**) observations; (**b**) the CMIP5 multi-model mean; (**c**) the CMIP5 simulation equivalent to +1 SD; (**d**) the median CMIP5 simulation; and (**e**) the CMIP5 simulation equivalent to −1 SD. The uncertainty ranges associated with the linear trends correspond to twice the standard error associated with the linear fit.

There seem to be broadly two schools-of-thought within the scientific community on the relevance of the multi-model means. Some researchers argue that the "internal variability" of the climate models is essentially "noise", and that by averaging together the results of multiple models you can improve the "signal-to-noise" ratio, (e.g., [1,32,44–47]). Others disagree and argue that this random noise is a "feature" of the models, which can somehow approximate the "internal variability" of nature, (e.g., [15–18,29,48–51]). Both camps agree that because the random fluctuations are different for each model run, they cancel each other out in the multi-model ensemble averages. Where they disagree is whether this is relevant for comparing model output to observations.

While we have demonstrated that the multi-model mean cannot fully explain the observed trends in annual snow cover, it is important to also consider the possibility that this is due to the lack of "internal variability" in the multi-model means. There are several methods to address this. For example, when comparing observed and modelled Arctic sea ice trends, Connolly et al. [1] considered both the multi-model mean and the median model run (in terms of long-term sea ice trends). Rupp et al. [15], by contrast, used the model output from "pre-industrial simulations" that were run without any "external forcing" to estimate the "internal variability", and Mudryk et al. [16] used an ensemble of 40 model runs that all used the same climate model and identical "external forcing". Other groups have calculated confidence intervals from the entire range of model output (e.g., the upper 5% and lower 5%) [17,18,29,45–51].

Here, we consider the "internal variability" of the models by analyzing three representative model runs. All model runs were ranked according to their 1967–2016 linear annual trend (Figure 2e). The mean and standard deviation was calculated for all 196 model runs. We then identified (i) the median model run and the model runs whose linear trends were closest to (ii) +1 standard deviation and (iii) −1 standard deviation (Figure 4c–e).

Comparing individual model runs to observations yields a more favorable comparison than using the multi-model mean. That is, the individual runs show more year-to-year variability than the multi-model mean. Nonetheless, the individual models still poorly explain the observed trends and all three selected models (i.e., the median and +/− one standard deviation) suggest a fairly continuous long-term decrease in snow cover extent.

If the lack of internal variability in the multi-model mean is proposed as the explanation for the discrepancies between the multi-model mean and the observations, then this does not vindicate the robustness of the climate models. Rather, it merely argues that the models are "not totally inconsistent with" the observations. This argument becomes weaker when the individual seasonal trends are examined.

The above analysis is repeated but for each of the seasonal averages instead of the annual averages—winter (Figure 5); spring (Figure 6); summer (Figure 7); and autumn (Figure 8). Note that the median, +1 standard deviation, and −1 standard deviation model runs for each of these seasons were not necessarily the same as for the annual averages.

First, consider the modelled winter (DJF) snow cover trends compared to observations (Figure 5). Climate models predicted a long-term decrease in winter snow cover, but this has not been observed. Indeed, the observations imply a net increase in winter snow cover, although this is not statistically significant. At any rate, since the start of the 21st century, snow cover has mostly been above the 1967–2016 average.

Collectively, climate models predicted a statistically significant decrease in winter snow cover which has not been observed (Figure 5c), even after more than fifty years of observations. However, for *some* of the models (e.g., the +1 standard deviation model), the modelled decrease is not statistically significant.

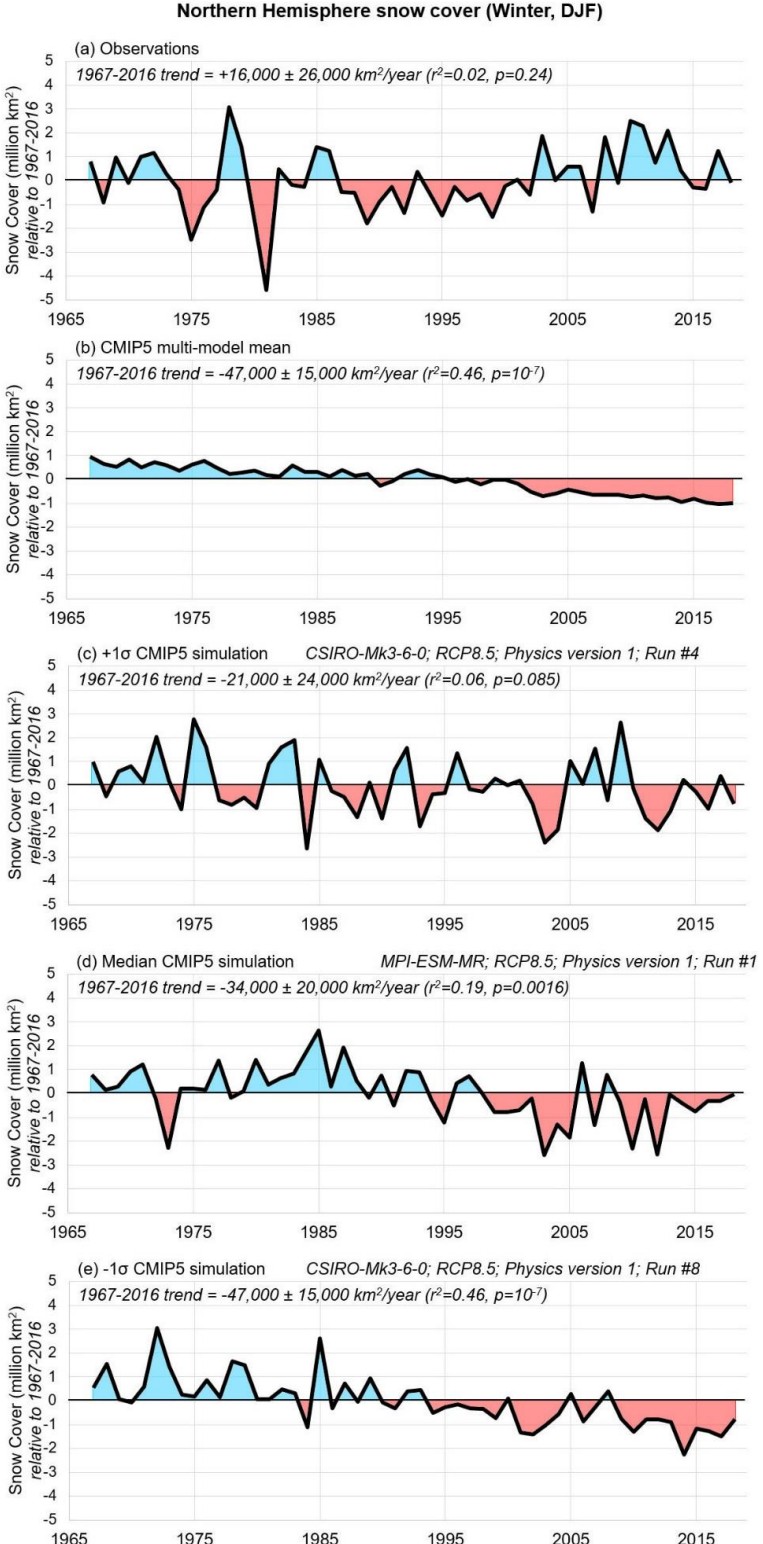

**Figure 5.** Same as Figure 4, except showing the winter (DJF, i.e., December, January and February).

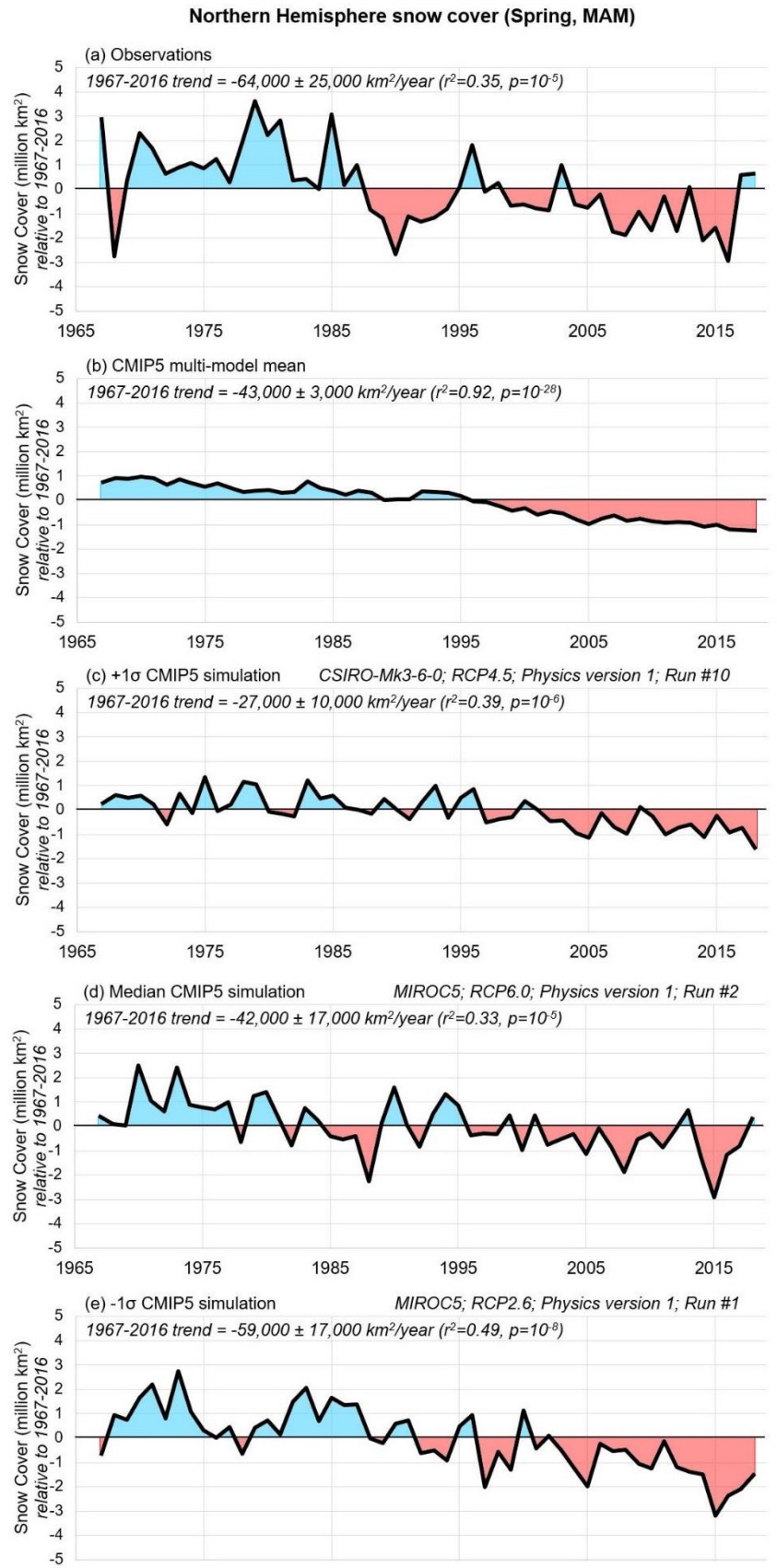

**Figure 6.** Same as Figure 4, except showing the spring (MAM, i.e., March, April and May).

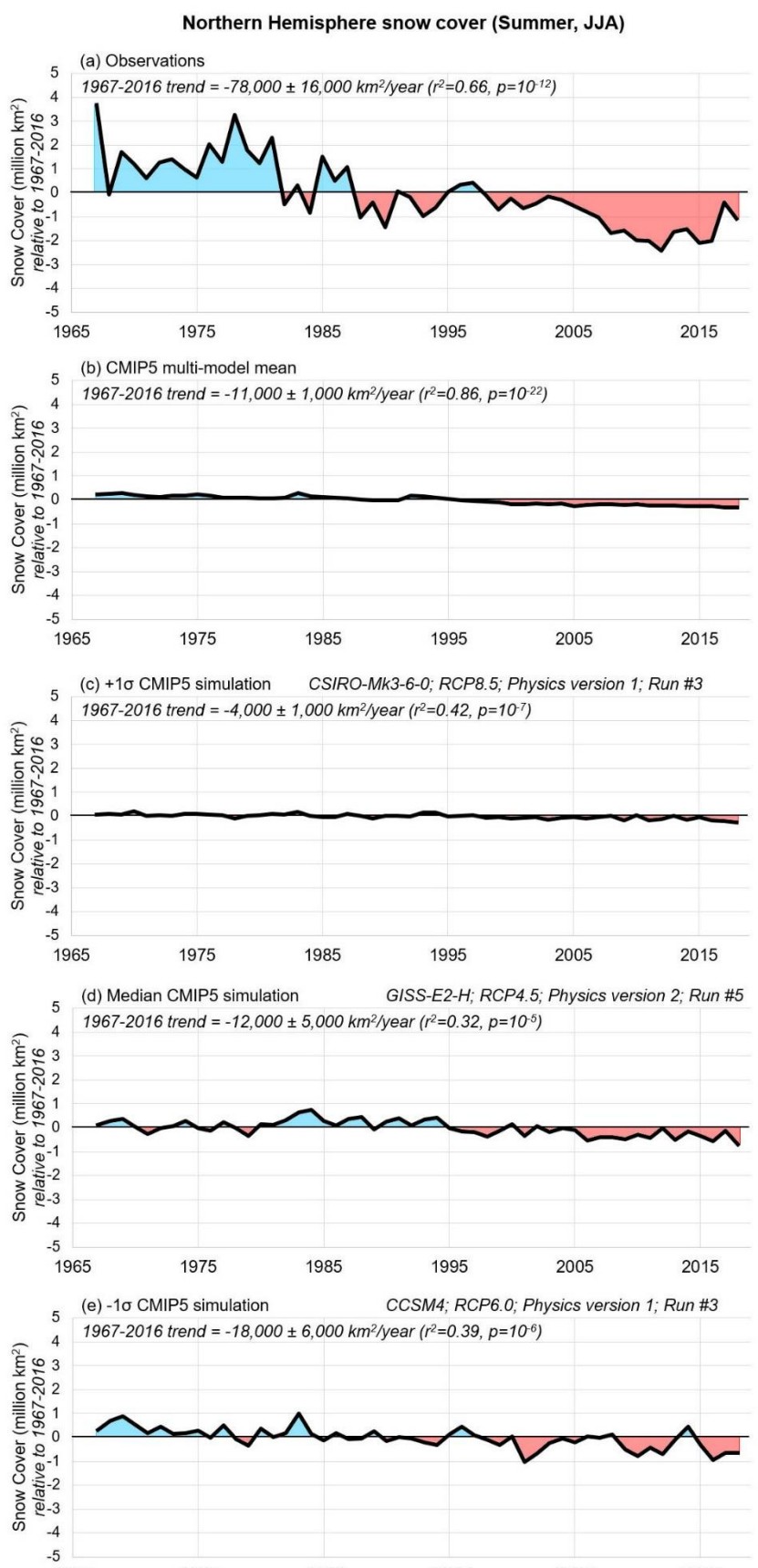

**Figure 7.** Same as Figure 4, except showing the summer (JJA, i.e., June, July and August).

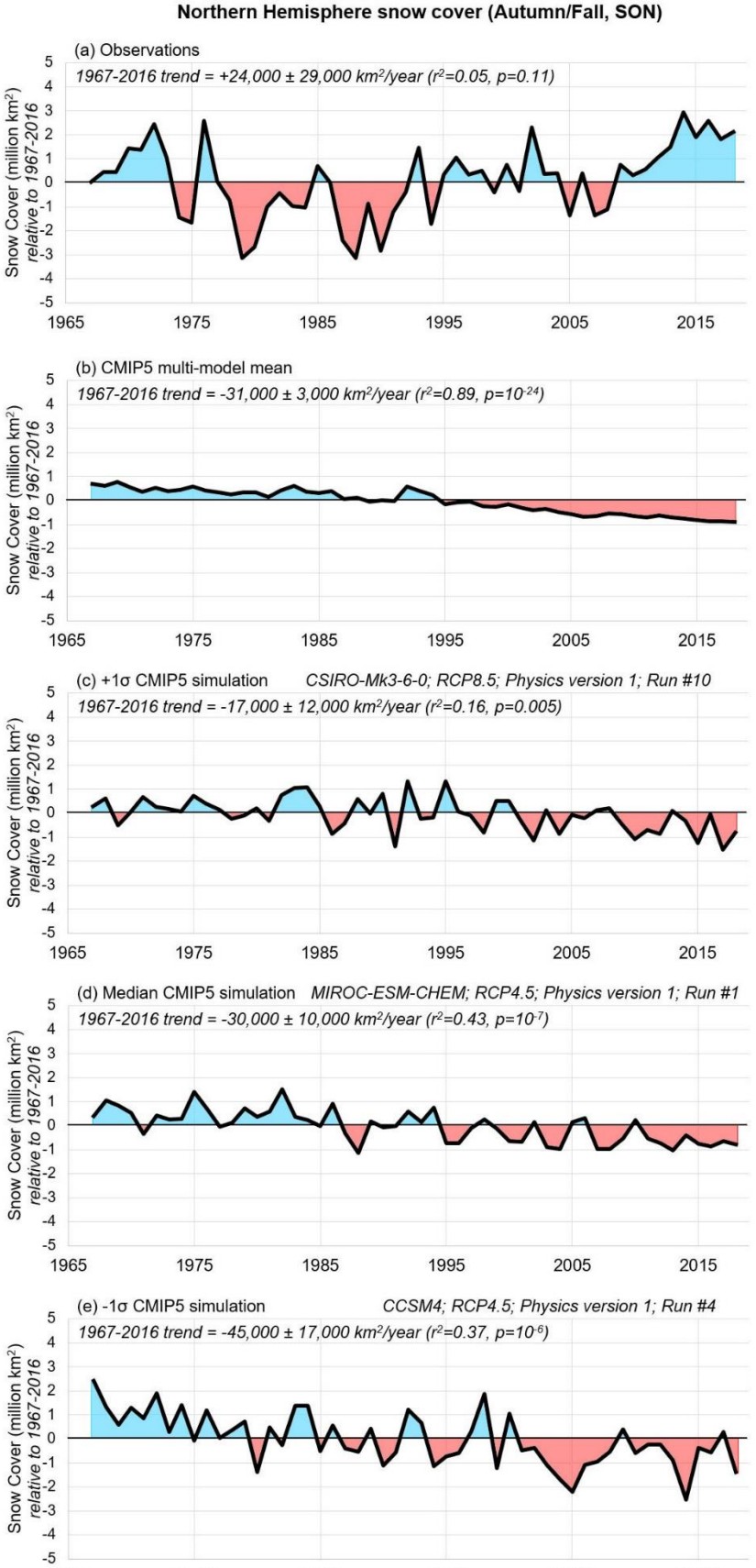

**Figure 8.** Same as Figure 4, except showing the autumn/fall (SON, i.e., September, October and November).

Results for spring (MAM—Figure 6) are more encouraging for the climate models, although notable discrepancies still exist between the modelled and observed trends. Perhaps this partially explains why this is the season which has received the most attention, (e.g., [11,13–18,29,30]).

Although the trends were all negative, the magnitude of the observed trend was greater than what most of the models had predicted—this can also be seen from Figure 3b. This has already been noted by others [11,14–18,30], although the typical implication is that the models performed well but simply "underestimated" the rate of the human-induced global warming to which the decrease is attributed. Derksen and Brown [30], for example, imply that the discrepancy is "… increasing evidence of an accelerating cryospheric response to global warming" ([30], p. 5).

Such an explanation is flawed. If the reason the models underestimated the negative trend in snow cover in spring (and summer) was because the models underestimated the effect of human-induced global warming, then their failure to explain winter and autumn is even more significant. Moreover, as previously noted, most of the decrease in spring snow cover occurred as a step-like behavior in the late-1980s [24–26], and the two most recent years (2017 and 2018) had values above the 1967–2016 average.

Like spring, modelled trends in summer (JJA—Figure 7) were negative, commensurate with the observed trends. However, this was where the similarities ended. The observed decrease in summer was greater than for spring, but the modelled decline was much more modest for summer. That is, the discrepancy between the modelled and observed trends was even greater for summer than spring—which was particularly striking (see Figure 3c). A partial explanation might be that the climate models significantly underestimated the total summer snow cover (see Figure 2c). However, the models poorly explained the observed summer trends.

Trends for autumn/fall (SON—Figure 8) were broadly similar to those for winter, but the contrast between the observed and modelled trends was even greater. Although the observed autumn snow cover decreased in the late-1970s, it had mostly been above the 1967–2016 average since the early-1990s (Figure 8a). As for the other seasons, all models implied an almost continuous decline in autumn snow cover which was not reproduced in the observations.

Brown and Derksen [40] suggest that the Rutgers dataset overestimated the October snow cover extent for Eurasia in recent years, which could partially explain some of the disagreement among the models [16,40]. However, we note that the Rutgers dataset was likely to be reasonably accurate because the weekly satellite-derived charts from which it was constructed, used operationally, and were manually evaluated by a human team for accuracy [27].

### 3.2. Comparison of CMIP5 Climate-Modelled March/April Trends to the Updated Brown and Robinson Time-Series (1922–2018)

Observed spring snow-cover trends for the Northern Hemisphere obtained from the updated Brown and Robinson time-series were compared to those of the climate models (Figure 9). Since the original time-series covers only the period 1922–2010 [28], trends for this period were only analyzed for this 88-year period, although all series were plotted to the most recent data point (i.e., 2018).

Results were similar to those of Figure 6. While all series implied a negative trend, the observations implied a greater decrease in snow cover than the models had predicted. Meanwhile, the pattern of the trends for the models were distinct from the observations. The models implied there should have been a gradual, but almost continuous decrease since the latter half of the 20th century, while the observed trends were more consistent with the step-like decrease in the late-1980s—as has already been noted by others [24–26]. The observed annual variability was quite substantial.

**Figure 9.** As for Figure 4, except for the updated Brown & Robinson (2011) March/April "Spring" estimate, and covering the period 1922–2018.

The strongly non-linear nature of the observed trends implies that reporting the data in terms of a "linear trend" (over some fixed period) is highly misleading. However, we note that this is essentially what the IPCC did in their 5th Assessment Report:

> "There is very high confidence that the extent of Northern Hemisphere snow cover has decreased since the mid-20th century (see Figure SPM.3). Northern Hemisphere snow cover extent decreased 1.6% [0.8 to 2.4%] per decade for March and April, and 11.7% [8.8 to 14.6%] per decade for June, over the 1967 to 2012 period. During this period, snow cover extent in the Northern Hemisphere did not show a statistically significant increase in any month." ([29], pp. 7–8)

Their Figure SPM.3 refers to a plot of the Brown and Robinson (2011) [28] March/April "spring" snow-cover time-series (apparently updated to 2012 using the Rutgers dataset). Based on these data, we agree that the Northern Hemisphere spring snow cover extent decreased over the 1967–2012 period. However, the data before and after that period (1967–2012, Figure 10) show a general *increase* in snow cover. In hindsight, the decision by the IPCC to emphasize the linear trends for such a specific period was unwise and considerably misleading.

We do not wish to read too much into the fact that the linear trends before and after the 1967–2012 are positive—indeed the trends are not statistically significant. Rather, we want to stress that the time-series is strongly non-linear and describing the series in terms of a single linear trend (over any time period) is inappropriate. An example of a more appropriate method of considering the non-linear nature of the time-series is shown in Figure 11.

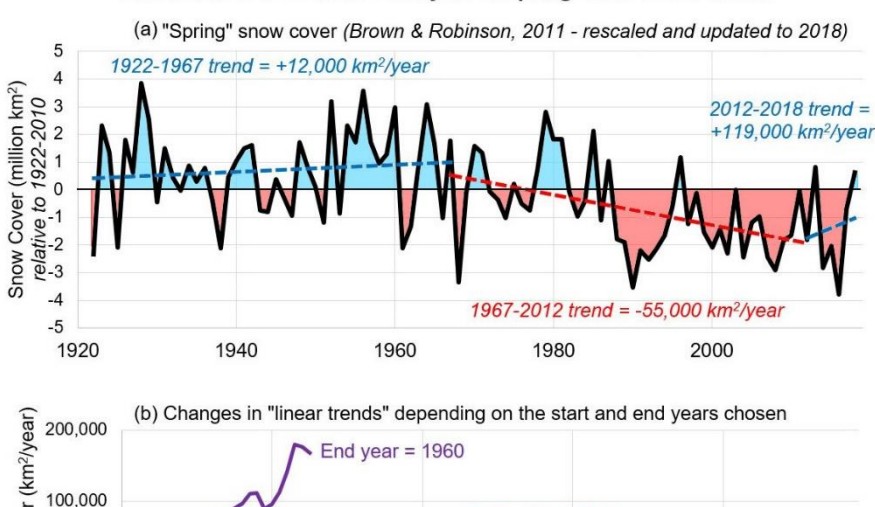

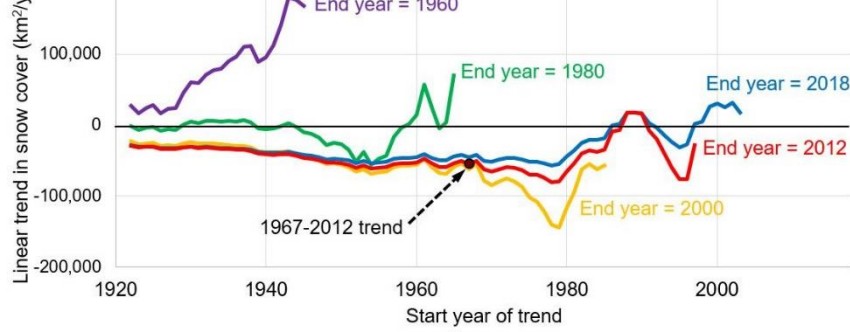

**Figure 10.** (**a**) Comparison of the 1967–2012 linear trend in Northern Hemisphere spring snow cover extent (red dashed line) discussed by the IPCC AR5 Working Group 1 report with the trends before and after that period (blue dashed lines). (**b**) How the "linear trend" changes depending on the start and end years of the time period chosen.

Figure 11 shows the time-frequency wavelet analysis of the spring snow cover from 1922–2018 using the algorithm recently introduced by Velasco et al. [52] and Soon et al. [53]. The result mainly illustrates the rich spectral content of the spring snow cover where primary modulation with principal periodicities at 23, 7, 4, and 2.4 years were detected. Such observed oscillations do not appear to be adequately accounted for by the CMIP5 models. We have already mentioned that the CMIP5 models neglected to consider any of the published high-solar variability estimates of total solar irradiance [32], so this could explain the poor performance of the models. We also note here the importance of considering the changes in short-term orbital forcing which have been of the order of 1–3 W/m$^2$ (depending on season) over the 20th century [54,55].

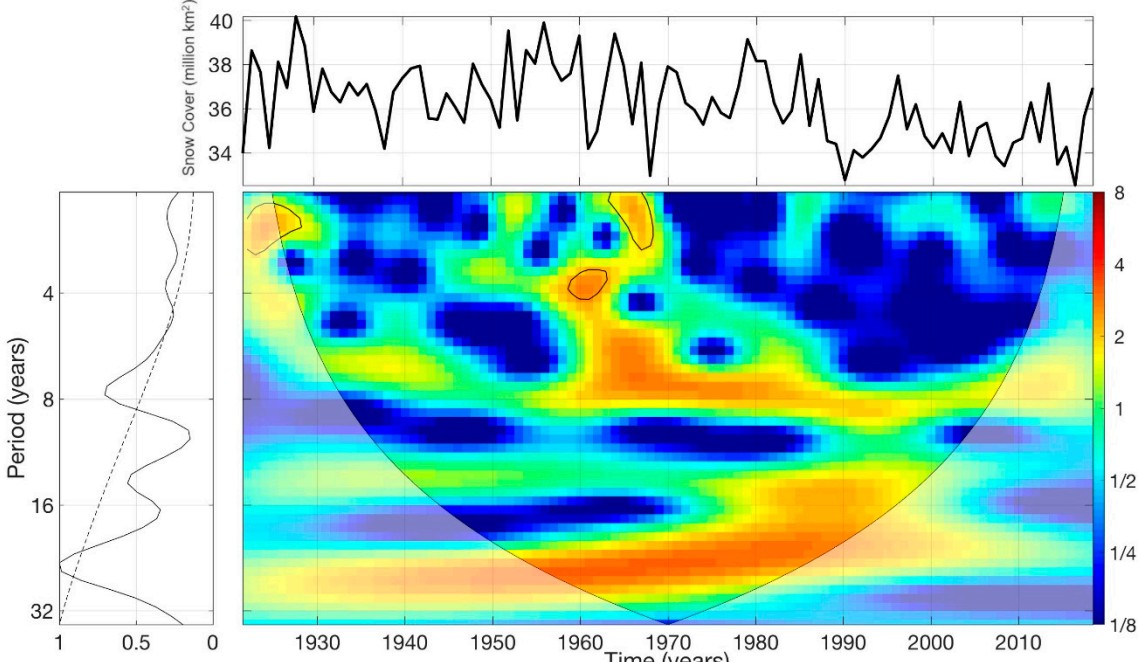

**Figure 11.** Time-frequency wavelet power (middle panel) of the Northern Hemisphere spring snow cover from 1922–2018. Top panel shows the original time-series, while the left panel shows the global time-averaged spectrum indicating significant (95% above the adopted reference red noise spectrum) oscillations at about 23, 7, 4, and 2.4 years.

## 4. Discussion and Conclusions

In this paper, the observed changes in Northern Hemisphere snow cover extent since 1967 were compared to the changes predicted by the CMIP5 climate models. In total, 196 climate model runs (taken from 24 climate models and 15 climate modelling groups) were analyzed. A longer time-series that was also available for Northern Hemisphere spring (March–April), beginning in 1922 [28], was also compared to the equivalent climate model predictions.

According to the climate models, snow cover should have steadily decreased for all four seasons. However, the observations show that only spring and summer demonstrated a long-term decrease. Indeed, the trends for autumn and winter suggest a long-term increase in snow cover, although these trends were not statistically significant. Moreover, the decrease in spring (and to a lesser extent, summer) was mostly a result of an almost step-like decrease in the late-1980s, which is quite different from the almost continuous gradual decline expected by the climate models.

The CMIP5 climate models expected the decline in snow cover across all seasons because they assume:

(1)　Northern Hemisphere snow cover trends are largely determined by their modelled global air temperature trends.

(2)  They contend that global temperature trends since the mid-20th century are dominated by a human-caused global warming from increasing atmospheric greenhouse gas concentrations [29].

The fact that the climate models expect snow cover trends to be dominated by a human-caused global warming is confirmed by the formal "detection and attribution" studies of spring snow-cover trends [15,17]. However, the inability of the climate models to accurately describe the observed snow cover trends indicates that one or both assumptions are problematic. Several possible explanations exist:

(a)  The models may be correct in their predictions of human-caused global warming, yet are missing key atmospheric circulation patterns or effects which could be influencing Northern Hemisphere snow-cover trends [36,37,41,42].

(b)  The models might be overestimating the magnitude of human-caused global warming, and thereby overestimating the "human-caused" contribution to snow-cover trends. This would be consistent with several recent studies which concluded that the "climate sensitivity" to greenhouse gases of the climate models is too high [56–58].

(c)  The models might be underestimating the role of natural climatic changes. For instance, the CMIP5 models significantly underestimate the naturally occurring multidecadal trends in Arctic sea ice extent [1]. Others have noted that the climate models are poor at explaining observed precipitation trends [47,48,59], and mid-to-upper atmosphere temperature trends [44–46].

(d)  The models might be misattributing natural climate changes to human-caused factors. Indeed, Soon et al. [32] showed that the CMIP5 models neglected to consider any high-solar variability estimates for their "natural forcings". If they had, much or all of the observed temperature trends could be explained in terms of changes in the solar output.

It is possible that more than one of the above factors is relevant, therefore we would encourage more research into each of these four possibilities. At any rate, for now, we recommend that the climate model projections of future and past snow-cover trends should be treated with considerable caution and skepticism. Changes in the Northern Hemisphere snow cover have important implications for society [6] and local ecosystems [7]. Therefore, it is important that people planning for future changes in snow cover do not rely on unreliable projections.

One short-cut which regional and global climate modellers could use to potentially improve the reliability of their snow cover projections is to apply "bias corrections" to bring the hindcasts more in line with observations. This is a technique which has now become a standard procedure in climate change impact studies, e.g., see Ehret et al. [60]. However, we agree with Ehret et al. [60] that any such bias corrections should be made clear and transparent to the end users.

In the meantime, more than 50 years of satellite data exist (Figure 4a, Figure 5a, Figure 6a, Figure 7a, and Figure 8a) to estimate the climatic variability in snow cover for each of the seasons, as well as nearly 100 years of data for spring snow cover (Figure 9a). Consequently, the observed historical variability for each of the seasons is a far more plausible starting point than the current climate model projections for climate change adaptation policies. With this in mind, we have provided the various time series and data used in this paper as Supplementary Materials.

**Supplementary Materials:** The various time series and data used for constructing Figures 1–9 are available online at http://www.mdpi.com/2076-3263/9/3/135/s1.

**Author Contributions:** R.C., M.C., and W.S. carried out most of the conceptualization, methodology, and formal analysis. R.C., M.C., W.S., D.R.L., R.G.C., and V.M.V.H. contributed to the writing of this article.

**Funding:** R.C. and W.S. received financial support from the Center for Environmental Research and Earth Sciences (CERES), http://ceres-science.com/, while carrying out the research for this paper. The aim of CERES is to promote open-minded and independent scientific inquiry. For this reason, donors to CERES are strictly required not to attempt to influence either the research directions or the findings of CERES.

**Acknowledgments:** We acknowledge the World Climate Research Program's Working Group on Coupled Modelling, which is responsible for CMIP, and we thank the climate modeling groups (listed in Table 1 of this paper) for producing and making available their model output. For CMIP, the US Department of Energy's Program for Climate Model Diagnosis and Intercomparison provided coordinating support and led development of software infrastructure in partnership with the Global Organization for Earth System Science Portals. We are grateful to Geert Jan van Oldenborgh for creating and maintaining the KNMI Climate Explorer website, https://climexp.knmi.nl/, which we used to obtain the CMIP5 hindcasts/projections data and to the Rutgers University Global Snow Lab, https://climate.rutgers.edu/snowcover/, for constructing and maintaining their Northern Hemisphere snow cover datasets.

**Conflicts of Interest:** The authors declare no conflict of interest. The funders had no role in the design of the study; in the collection, analyses, or interpretation of data; in the writing of the manuscript, or in the decision to publish the results.

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
