# Peer review of "Northern Hemisphere Snow-Cover Trends (1967–2018): A Comparison between Climate Models and Observations"

_geosciences, doi:10.3390/geosciences9030135_

Round 1

Reviewer 1 Report

This is a good article, clearly presented and easy to read. The introduction provides the relevant background information in a clear and detailed way. Methods are clearly presented. The results are rather sound, and the discussion adequately presents potential explanations for the poor performance of global climate models in representing recent snow cover trends. The comparison of model snow cover trends with observational trends is not a novel approach per se, but, as the authors clearly state, such comparison on a seasonal basis and their discussion on the validity of using linear trends for such comparison is scientifically interesting and original.

Main comments:

- Some formulations are a bit too "politically" or "philosophically engaged" for a scientific article. These include, for example, p. 2, lines 64-65 "the human-caused global warming theory", p.2 lines 94-96, "(...) as part of its justification for its claim that... (refering to the IPCC AR5)", or p. 13 lines 344-345 "supporters of the climate models". Please rephrase to make it sound less controversial (or almost skeptical about the human influence on climate).

- The authors are very critical regarding climate models. Their analysis indeed shows major (well-known) weaknesses of models compared to observations (poor representation of trends, too low interannual variability, ...). However, the observational uncertainty is rarely discussed. I think this should be further emphasised in order to provide a fair comparison between models and observations. Please see my specific comments below for some suggestions of improvement concerning this.

- Concerning the last paragraph of your conclusions, I agree with you in the sense that climate projections are still far from reliable. However, if you want to look into the future, model outputs will be necessary at some point (as you correctly point out, the observed historical variability is only a starting point). It is a well-known thing in the snow modelling community that land surface schemes used in global climate models poorly represent snow, mainly because of their low complexity (usually not physics-based, lacking some processes, etc.). For more regional to local assessment of snow cover, the "impact" community usually uses the GCMs or GCM-fed RCMs to force their own snow models, that rely on temperature, precipitation, but also radiative fields, humidity, etc from the climate models. Snow cover results from these model chains are usually much closer to observations than the ones coming directly from CMIP5 GCMs (especially when bias adjustment of the climate model outputs is performed beforehand). This is probably something worth mentioning somewhere in the discussion or conclusions. Also, for the short-to-medium term climate change adaptation policies, it might be relevant to have a look at decadal prediction model outputs, which should be more reliable since they are initialised with observations. Have you considered this? Also, for longer term, using bias-corrected climate projections might be another option (also for the regional assessments using more complex snow models). Those are some emerging fields of research that might be worth mentioning here, that would give a broader perspective concerning snow in climate models in your conclusions.

Specific comments:

- p. 1, line 33: reference [3] might not be the most appropriate for such a general statement about ice sheets and glaciers, as it refers to glaciers from the tropical Andes only. Please use a more global reference (maybe from IPCC?).

- p. 1, lines 34-35: please rephrase. The way the sentence is formulated, it seems as if you're stating that snowmelt (and not snow) supports a large winter outdoor recreation industry...

- p. 2, line 81: the word "that" should be removed, i.e. "rule out the possibility that..."

- p. 4, lines 166-175 and Table 1: here it's not clearly stated that you use the historical-RCP runs as "continuous" runs (historical until 2005, then RCP 2.6, 4.5, 6.0 or 8.5). Please include this information.

- p. 4, lines 173-175. The fact that "the differences between the projections by 2018 are relatively modest", would motivate the use of a single RCP rather than treating "each of the scenario
runs as a separate run". Did you check how different the model runs for different RCPs are at the end of your analysis period (in 2016 or 2018) and if this motivates the use of the 4 different RCPs for this recent period?

- p. 5 lines 201-203 and Fig. 1b): you mention error bars here. A way of emphasisingobservational uncertainty might be to put those error bars in Figure 1 b). Also the Rutgers dataset probably comes with some estimates of uncertainty that should be indicated somewhere (for example in this figure).

- Figures 2-3: the observational range of uncertainty should be added in these figures as well, for example in the written value, i.e. "Observed 46 ± X million km2". The legends also lack the explanation about the dashed red lines (model average).

- p. 11, line 294: there's a typo here, an "s" is missing in "There seems to be".

- p. 11, lines 336-339. The reference to Figure 5 c) seems to be in the wrong place. It should probably be indicated at the end of the paragraph, line 339.

- p. 13, lines 354-357: I don't see how the sentence starting with "Moreover," further explains that the explanation presented in the previous paragraph is flawed?

- p. 13, lines 372-374: do you have some quantified uncertainty estimate to support your assumption that "the Rutgers dataset is likely to be reasonably accurate"?

- p. 17, line 410-411 and Figure 10: performing a trend on a period of less than 10 years, for instance 2012-2018, does not make any sense climatologically nor statistically. Please remove this part.

- p. 17, line 421: there seems to be a verb missing here. Did you mean "where primary modulation with principal periodicities at 23, 7, 4 and 2.4 years are detected"?

- p. 19, line 445: the "s" in "demonstrates" should be removed ("spring and summer demonstrate").

- I found it really nice that the data used in this study is available as a supplement. However, if I'm not mistaken, this is never mentioned in the article. It should be indicated somewhere.

Author Response

Response to Reviewer 1 Comments

Point 1: This is a good article, clearly presented and easy to read. The introduction provides the relevant background information in a clear and detailed way. Methods are clearly presented. The results are rather sound, and the discussion adequately presents potential explanations for the poor performance of global climate models in representing recent snow cover trends. The comparison of model snow cover trends with observational trends is not a novel approach per se, but, as the authors clearly state, such comparison on a seasonal basis and their discussion on the validity of using linear trends for such comparison is scientifically interesting and original.

Response 1: Thank you for your encouraging and helpful comments

Main comments:

Point 2: - Some formulations are a bit too "politically" or "philosophically engaged" for a scientific article. These include, for example, p. 2, lines 64-65 "the human-caused global warming theory", p.2 lines 94-96, "(...) as part of its justification for its claim that... (refering to the IPCC AR5)", or p. 13 lines 344-345 "supporters of the climate models". Please rephrase to make it sound less controversial (or almost skeptical about the human influence on climate).

Response 2: Fixed.

Point 3: - The authors are very critical regarding climate models. Their analysis indeed shows major (well-known) weaknesses of models compared to observations (poor representation of trends, too low interannual variability, ...). However, the observational uncertainty is rarely discussed. I think this should be further emphasised in order to provide a fair comparison between models and observations. Please see my specific comments below for some suggestions of improvement concerning this.

Response 3: Good point. While Brown & Robinson (2011) includes uncertainty estimates, the Rutgers dataset does not. We have modified the text to highlight this fact – see discussion in Response 10 below.

Point 4: - Concerning the last paragraph of your conclusions, I agree with you in the sense that climate projections are still far from reliable. However, if you want to look into the future, model outputs will be necessary at some point (as you correctly point out, the observed historical variability is only a starting point). It is a well-known thing in the snow modelling community that land surface schemes used in global climate models poorly represent snow, mainly because of their low complexity (usually not physics-based, lacking some processes, etc.). For more regional to local assessment of snow cover, the "impact" community usually uses the GCMs or GCM-fed RCMs to force their own snow models, that rely on temperature, precipitation, but also radiative fields, humidity, etc from the climate models. Snow cover results from these model chains are usually much closer to observations than the ones coming directly from CMIP5 GCMs (especially when bias adjustment of the climate model outputs is performed beforehand). This is probably something worth mentioning somewhere in the discussion or conclusions. Also, for the short-to-medium term climate change adaptation policies, it might be relevant to have a look at decadal prediction model outputs, which should be more reliable since they are initialised with observations. Have you considered this? Also, for longer term, using bias-corrected climate projections might be another option (also for the regional assessments using more complex snow models). Those are some emerging fields of research that might be worth mentioning here, that would give a broader perspective concerning snow in climate models in your conclusions.

Response 4: These are very useful suggestions, and we have added some extra comments along those lines to the conclusions, as well as a citation to the Ehret et al. (2012) paper which provides some useful discussion on many of these points.

Specific comments:

Point 5: - p. 1, line 33: reference [3] might not be the most appropriate for such a general statement about ice sheets and glaciers, as it refers to glaciers from the tropical Andes only. Please use a more global reference (maybe from IPCC?).

Response 5: This is a fair point. We have added a second reference, Zemp et al. (2009) which reviews the worldwide glacier monitoring network from a mass-balance perspective.

Point 6: - p. 1, lines 34-35: please rephrase. The way the sentence is formulated, it seems as if you're stating that snowmelt (and not snow) supports a large winter outdoor recreation industry...

Response 6: Fixed.

Point 7: - p. 2, line 81: the word "that" should be removed, i.e. "rule out the possibility that..."

Response 7: Fixed.

Point 8: - p. 4, lines 166-175 and Table 1: here it's not clearly stated that you use the historical-RCP runs as "continuous" runs (historical until 2005, then RCP 2.6, 4.5, 6.0 or 8.5). Please include this information.

Response 8: Thanks, we agree this could potentially have been confusing. Fixed

Point 9: - p. 4, lines 173-175. The fact that "the differences between the projections by 2018 are relatively modest", would motivate the use of a single RCP rather than treating "each of the scenario
runs as a separate run". Did you check how different the model runs for different RCPs are at the end of your analysis period (in 2016 or 2018) and if this motivates the use of the 4 different RCPs for this recent period?

Response 9: We’ve added a brief discussion of this now.

Point 10: - p. 5 lines 201-203 and Fig. 1b): you mention error bars here. A way of emphasisingobservational uncertainty might be to put those error bars in Figure 1 b). Also the Rutgers dataset probably comes with some estimates of uncertainty that should be indicated somewhere (for example in this figure).

Response 10: Brown & Robinson (2011) calculated estimates of the uncertainty for their March/April time series, but the Rutgers dataset doesn’t actually provide any estimates. We’ve added some extra discussion to clarify this.

Point 11: - Figures 2-3: the observational range of uncertainty should be added in these figures as well, for example in the written value, i.e. "Observed 46 ± X million km2". The legends also lack the explanation about the dashed red lines (model average).

Response 11: We had originally included the confidence intervals for the trends there as suggested, but we found that the plots became too cluttered as a result. We do include these in Figures 4-9, however. We have added an explanation about the dashed red lines, as suggested.

Point 12: - p. 11, line 294: there's a typo here, an "s" is missing in "There seems to be".

Response 12: Fixed.

Point 13: - p. 11, lines 336-339. The reference to Figure 5 c) seems to be in the wrong place. It should probably be indicated at the end of the paragraph, line 339.

Response 13: Fixed.

Point 14: - p. 13, lines 354-357: I don't see how the sentence starting with "Moreover," further explains that the explanation presented in the previous paragraph is flawed?

Response 14: We have added an extra discussion to elaborate on this.

Point 15: - p. 13, lines 372-374: do you have some quantified uncertainty estimate to support your assumption that "the Rutgers dataset is likely to be reasonably accurate"?

Response 15: This is based on the discussion in the Frei et al. (2012) review of the relevant satellite datasets and references therein. We have slightly modified the discussion to clarify this.

Point 16: - p. 17, line 410-411 and Figure 10: performing a trend on a period of less than 10 years, for instance 2012-2018, does not make any sense climatologically nor statistically. Please remove this part.

Response 16: We had been making essentially the same point and had explicitly stated that the “linear trends” both before and after the 1967-2012 were not statistically significant. But, we appreciate that our original discussion might potentially have been misinterpreted. For that reason, we have modified this discussion and added a brief extra analysis (Figure 10b) to make this clearer.

Point 17: - p. 17, line 421: there seems to be a verb missing here. Did you mean "where primary modulation with principal periodicities at 23, 7, 4 and 2.4 years are detected"?

Response 17: Fixed.

Point 18: - p. 19, line 445: the "s" in "demonstrates" should be removed ("spring and summer demonstrate").

Response 18: Fixed.

Point 19: - I found it really nice that the data used in this study is available as a supplement. However, if I'm not mistaken, this is never mentioned in the article. It should be indicated somewhere.

Response 19: Thanks, we hope that it will be of value to the scientific community. We have added a reference to the fact that the data is available (using the journal template).

Reviewer 2 Report

General comments

The paper entitled “Northern Hemisphere snow cover trends (1967-2018): A comparison between climate models and observations” by Connolly et al., deals with the comparison between observed changes in Northern Hemisphere sown cover from satellites record and those predicted by 196 available run by CMIP5 climate models over a period of about 50 years.

The paper is well written and the introduction is well documented by a robust literature. The work presents interesting results that deserved to be published after minor revisions.

Specific comments

P1, L17: I suggest: “…compared to those predicted by all available CMIP5 climate models…”

Please, check some words written in bold that appear in the text, like P1-L43, P11-L332, etc.

Figure 1a: why do plot in x-axis the Brown & Robison time series before rescaling and not after rescaling as in Figure 1b?

P7, L230: I think it 196 CIMP5

Author Response

Response to Reviewer 2 Comments

Point 1: General comments 

The paper entitled “Northern Hemisphere snow cover trends (1967-2018): A comparison between climate models and observations” by Connolly et al., deals with the comparison between observed changes in Northern Hemisphere sown cover from satellites record and those predicted by 196 available run by CMIP5 climate models over a period of about 50 years.

The paper is well written and the introduction is well documented by a robust literature. The work presents interesting results that deserved to be published after minor revisions.

Response 1: Thank you for your encouraging and helpful comments.

Specific comments

Point 2:  P1, L17: I suggest: “…compared to those predicted by all available CMIP5 climate models…”

Response 2: Done.

Point 3: Please, check some words written in bold that appear in the text, like P1-L43, P11-L332, etc.

Response 3: We originally used bold in four spots to emphasise a particular word, but we take the reviewers point and have now removed the bolding, although we used italics to keep the emphasis in two cases.

Point 4: Figure 1a: why do plot in x-axis the Brown & Robison time series before rescaling and not after rescaling as in Figure 1b?

Response 4: The main point of Figure 1a was to show that the relationship between the two datasets was almost exactly linear during the period of overlap, and to derive the rescaling factors which were the slope and intercept of the linear least squares fit [shown in the grey shaded box]. We have slightly modified the text to make this clearer.

Point 5: P7, L230: I think it 196 CIMP5

Response 5: Yes, thanks!